# Air Quality Geospatial Analysis in Vulnerable Areas. Case Study of Valencia (Spain)

**DOI:** 10.3390/ijerph21101278

**Published:** 2024-09-25

**Authors:** Nuria Guardiola Ibáñez, Eloina Coll Aliaga, Maria Joaquina Porres De La Haza, Victoria Lerma Arce, Edgar Lorenzo-Sáez

**Affiliations:** 1Universitat Politècnica de Valencia, 46022 Valencia, Spain; ecoll@cgf.upv.es (E.C.A.); mporres@cgf.upv.es (M.J.P.D.L.H.); 2ITACA Research Institute, Universitat Politècnica de Valencia, Camí de Vera s/n, 46022 Valencia, Spain; vlerma@upv.es (V.L.A.); edlosae@etsiamn.upv.es (E.L.-S.)

**Keywords:** nitrogen dioxide, air quality, vulnerability, passive dosimetry, environmental equality, geospatial analysis

## Abstract

The escalating concern over poor air quality, particularly nitrogen dioxide (NO_2_), poses a critical public health challenge, especially for vulnerable populations, such as children, older adults, and those with chronic diseases. This study aimed to analyze air quality in areas with vulnerable populations through a geospatial analysis of NO_2_ concentration measured by the passive dosimetry method in 2022. The results reveal high vulnerability caused by areas with over-centralized facilities and high concentrations of nitrogen dioxide, often coinciding with busy avenues. The study emphasizes the urgent need to address air quality disparities, providing crucial insights for public decision-makers to allocate resources effectively and reduce environmental inequalities in the city, ultimately safeguarding the health of at-risk communities.

## 1. Introduction

Exposure to poor air quality represents a significant environmental challenge with profound implications for human health. According to the World Health Organization (WHO), in 2019, an alarming 99% of the global population lived in regions where air pollution levels exceeded air quality guidelines [1,2]. In terms of its effects on human health, poor air quality contributes to a variety of diseases and results in approximately 6.5 million individuals contracting illnesses attributable to air pollution annually [1]. This figure underscores the urgent need for comprehensive strategies to address and mitigate the effects of air pollution on a global scale.

Poor air quality is related to various atmospheric pollutants, with nitrogen dioxide (NO_2_) being one of the main contributors. This pollutant not only directly affects air quality, but also reacts with other pollutants, creating high correlations with various harmful substances, thus establishing high correlations with various pollutants [3]. This and particulate matter with an aerodynamic diameter less than or equal to 10 µm (PM) are the pollutants with the most significative negative impact on human health [4]. These pollutants are linked to high mortality rates due to prolonged exposure, which leads to respiratory diseases [5]. The interplay between NO_2_ and PM_10_ exacerbates the health risks, making it imperative to understand their sources and effects comprehensively.

Nitrogen dioxide (NO_2_) is a gas generated mainly during the combustion of fossil fuels and is present in vehicle engines, power plants, and industries. This air pollutant degrades air quality and adversely affects human health, causing eye, throat, and lung irritation, and exacerbating respiratory conditions. Nitrogen dioxide is responsible for 75,000 premature deaths in the European Union, according to the European Court of Auditors in 2018 [6].

Most nitrogen dioxide emissions are due to road traffic and a significant part of industrial activities, making it a predominantly anthropogenic pollutant. Consequently, the highest nitrogen dioxide concentrations are found in the most populated areas, especially in urban areas. In the future, urban or metropolitan regions will be the most densely populated, thereby becoming the most vulnerable areas. This trend necessitates proactive urban planning and policy measures to ensure sustainable and healthy living environments.

Despite the adverse health effects affecting the entire population due to NO_2_ exposure, there are four vulnerable population groups identified in the National Air Plan 2017–2019 [4]:1.Children aged 0–16: Prolonged exposure to poor air quality, particularly NO_2_, can lead to lung function deficits in school-aged children [7] and may even be associated with childhood stunting [8]. Protecting this group is crucial for ensuring their healthy development and long-term well-being.2.People over 65 years old: This age group is considered at risk due to the natural deterioration of their organs and their increased vulnerability to various diseases. Significant links have been observed in particular with nitrogen dioxide and the deterioration of asthma in older people suffering from this disease [9]. Ensuring air quality standards for this demographic is crucial for enhancing their quality of life.3.People with chronic diseases: Poor air quality can worsen health conditions and cause adverse effects [10] in people with cardiovascular problems, such as angina pectoris or arteriosclerosis, pregnant women, diabetics, and people who are overweight or obese, and especially in people suffering from chronic respiratory diseases, such as asthma or chronic obstructive pulmonary disease (COPD). Tailored healthcare strategies are essential to mitigate these risks.4.People who spend much time practicing outdoor sports: This population group has three of the four factors highlighted by the Ministry of Health: intense physical activity, in this particular case, with residence in urban areas and prolonged exposure to the open air, as long-term exposure to traffic-based pollution causes inflammation of the airways and lung function [11].

The main objective of this study was to conduct a geospatial analysis of air quality by measuring nitrogen dioxide levels in specific areas, with a focus on identifying vulnerable zones due to the susceptibility to this pollutant of their frequenters. Additionally, this study aimed to develop a methodology for this analysis that can be applied universally to any city. This methodological approach seeks to achieve environmental equity, ensuring that all populations, regardless of their geographical location, have healthier atmospheric conditions, thereby minimizing the risk to specific groups of people. By correlating data on NO_2_ concentrations with information on the location of vulnerable individuals, this study aimed to pinpoint critical areas within the study area that require special attention and the implementation of specific measures to improve air quality. This comprehensive approach will help eliminate environmental disparities and actively contribute to environmental equity by ensuring that all categories of the population, regardless of their particular characteristics, have a level playing field in their environment. By identifying and addressing these critical areas, policymakers can develop targeted interventions to reduce exposure and improve overall public health.

## 2. Materials and Methods

### 2.1. Study Area: Valencia

The study area focuses on the city of Valencia, located on the east coast of the Iberian Peninsula. This urban center plays the role of the capital at both the provincial and regional levels, being the head of the Valencian Community in Spain. Valencia’s strategic location along the Mediterranean coastline further emphasizes its significance in both historical and contemporary contexts, contributing to its dynamic economic and cultural landscape.

Valencia is ideal for this study due to its demographic and urban relevance in Spain. As one of the largest cities in the country, Valencia has a high population density, which implies a higher exposure to air pollution, particularly nitrogen dioxide (NO_2_), from road traffic. This high concentration of population and vehicles contributes significantly to NO_2_ levels, affecting many inhabitants. Additionally, Valencia’s diverse economic activities and dense urban infrastructure exacerbate the challenges associated with air quality management. According to previous studies, NO_2_ levels in Valencia are significantly above the safety levels proposed by the World Health Organization, affecting more than 99% of the city’s population [12].

Moreover, Valencia, located in the heart of the Valencian Community, acts as a receptor nucleus for vehicular emissions from surrounding municipalities [13]. The configuration and geographical distribution of the city favor the concentration of road traffic emissions from other localities, thus increasing the pollution load in its atmosphere. According to Catalin Ioan [13], Valencia is the primary recipient of these regional emissions, exacerbating air quality problems and their effects on public health. This geographic interplay underscores the complex dynamics of regional air pollution and necessitates coordinated efforts across municipal boundaries to effectively tackle the issue.

The choice of Valencia as the study area is further justified by several unique characteristics, such as the density of the population in the city and its mix of residential, commercial, and industrial zones. This diversity provides an ideal setting for examining the spatial distribution of NO_2_ concentrations. The city has significant traffic density, which is a major source of NO_2_ emissions. Additionally, Valencia’s demographic composition includes a mix of socioeconomic statuses, with certain neighborhoods housing more vulnerable populations. This variability allows for a detailed analysis of how NO_2_ pollution affects different demographic groups, particularly those more susceptible to respiratory issues. The extensive road network and heavy vehicle use make Valencia a critical area for studying traffic-related air pollution.

Figure 1 shows the 17 districts that have been included in the analysis (purple polygons in Figure 1), accounting for a land area of approximately 56 km^2^, with the exception of two districts, Poblats Nord and Poblats Sud (orange polygons in Figure 1), due to data availability limitations or data scarcity in the passive dosimetry campaign, which did not allow enough map continuity to perform the required analysis. This exclusion highlights the challenges in data collection and the importance of comprehensive monitoring systems to ensure accurate assessments.

### 2.2. NO_2_ Measurements

Passive dosimetry is a method to measure exposure to nitrogen dioxide concentrations. It counts with gas recording devices (sensors) that register NO_2_ concentration in a specific area over a specified period. This method is particularly advantageous due to its cost-effectiveness and ease of deployment in various environments, making it suitable for large-scale monitoring projects. The sensors used in this study were Palmes diffusion tubes, known for their reliability and accuracy in measuring environmental pollutants. Passive dosimetry sensors are characterized by their high reliability in measuring environmental pollutants. These sensors, which do not require an external power source to operate, are widely used in air quality studies due to their ability to provide accurate and consistent data over time. Their robust design and ability to integrate into various environments contribute to their effectiveness in capturing levels of exposure to contaminants. Additionally, their low maintenance requirements and ease of use make them ideal for long-term applications, allowing for the continuous collection of data that is essential for air quality analysis and modeling. The reliability of passive dosimetry sensors is supported by numerous studies demonstrating their accuracy and consistency in field conditions, making them an essential tool for environmental monitoring.

This study, implemented in 2022, comprised the deployment of passive dosimetry in specific locations over the study area in quarterly campaigns of 14-day samplings each, where NO_2_ concentration was recorded trying to avoid meteorological biases. To minimize meteorological biases, the measurements were conducted in different seasons and weather conditions, capturing a more comprehensive assessment of NO_2_ exposure levels. This allowed an evaluation of the cumulative or long-term exposure to NO_2_ (µg/m^3^) of the studied area. The measurements for the year 2022 were made in four different date ranges, all in 14-day campaigns, representing all stations and avoiding the meteorological biases inherent to each. This approach normalizes the data, ensuring that seasonal variations are accounted for, and by allowing for annual measures to be accurately determined by conducting multiple sampling campaigns throughout the year, the study aimed to capture seasonal variations in NO_2_ levels, providing a more comprehensive understanding of exposure patterns.

A total of 99 sensor stations were deployed in the study area, as can be seen in Figure 2: 17 sensors were placed in schools, another seven sensors were placed next to the official stations of the Valencian Network for Monitoring and Control of Air Pollution (RVVCCA), and the remaining ones were located throughout the city to represent it as much as possible. The criteria for selecting the sensor locations followed the macro- and micro-implementation principles established by the European Directive 2008/50/EC. 

The use of passive dosimetry for measuring NO_2_ concentrations in this study is validated through its comparison with the annual averages obtained from fixed air quality monitoring stations. As illustrated in Figure 3, the passive dosimeters’ measurements closely align with those of the continuous monitors, demonstrating that this method provides a reliable estimate of annual pollutant levels. Furthermore, passive dosimeters allow for greater spatial resolution, as they can be deployed in multiple locations across urban areas, providing a more detailed representation of local NO_2_ variations that fixed stations might miss.

### 2.3. Vulnerable Facilities

Considering the population groups most vulnerable to nitrogen dioxide (NO_2_) exposure, several types of facilities were selected for the study. The selection of these facilities is aligned with population groups in the framework outlined by the National Air Plan 2017–2019 [14].

Children aged 0–16 years: Daycare centers (0–3 years old) and Early Childhood Education Centers (0–6 years old).

Early Childhood Education centers (0–6 years old).Primary and secondary schools (for children aged 6–12 and 12–16 years old, respectively).Educational institutions and social centers that are oriented towards various age groups.

People over 65 years old: Residential homes: refers to specialized accommodation facilities for old people.Day centers for old people.Pensioners’ clubs and other meeting associations.

Chronically ill persons: Hospitals: medical institutions specializing in the care and treatment of patients.Health centers, including medical facilities that provide primary and preventive care services.Medical centers and clinics: refers to smaller medical facilities that provide specialized services.

People who spend a significant amount of time playing sports outdoors:Sports facilities with outdoor courts.Children’s playgrounds located in urban parks or protected areas of more than 1000 square meters. Includes outdoor play areas located in urban parks or large protected areas.

## 3. Data Treatment

The overall data treatment methodology was systematically organized into two main strands. The first strand involved the collection of nitrogen dioxide (NO_2_) data through passive dosimetry, followed by subsequent analysis and processing. Secondly, facilities were evaluated in relation to the equipment according to the different vulnerable groups and their geographical distribution. The data obtained from both strands were integrated into a comprehensive analysis to correlate these factors and derive pertinent conclusions regarding the impact of NO_2_ concentrations on vulnerable populations.

### 3.1. NO_2_ Concentration 

A total of 396 NO_2_ concentration measurements were acquired from the quarterly passive dosimetry campaigns. These data, which incorporate graphical and alphanumeric information, were processed through a series of processing operations.

Initially, the dataset was reprojected to the EPSG:25830/UTM zone 30N reference system to ensure spatial accuracy. Subsequently, a topological correction was carried out to eliminate the duplicated geometries derived from considering four annual measurements at each sensor site.

Regarding the alphanumeric information, null measurements were discarded and corrected, assigning a unique identifier to the stations that appeared duplicated because they had been used in different projects, calling them by different names in each project. An annual mean concentration was calculated using statistical techniques and assigned to each station. This information was exported to a geospatial format (vector points).

To create a continuous surface for pollutant distribution analysis, an interpolation was conducted between sensor stations using the inverse distance weighted (IDW) method. This method is very effective in generating continuous surfaces for pollutant distribution analyses due to its simplicity and ability to reflect local variations in data accurately. The IDW method was chosen for the following reasons:Ease of Implementation: IDW is straightforward to implement and computationally efficient, making it suitable for large datasets such as ours.Local Accuracy: IDW assigns greater weights to closer points, which is crucial for accurately capturing the spatial variability of NO_2_ concentrations, particularly in urban areas with dense monitoring networks.Parameter Selection: A distance coefficient (power parameter) of 2 was used, which balances the influence of nearby points without overly diminishing the impact of further points. This parameter was selected based on empirical testing and validation with known NO_2_ concentration patterns in the study area.

Other interpolation methods, such as Kriging, were considered. However, Kriging requires more complex assumptions about the data distribution and spatial autocorrelation, which may not necessarily improve the interpolation results given the density and distribution of our monitoring stations. The result of the interpolation was a raster layer, which was subjected to two different classifications (percentiles and World Health Organization limits) to facilitate the visualization of the data according to various criteria of interest:The first classification applied to the resulting raster layer, by percentiles, is used to group the areas homogeneously with low, medium, and high values [15]. Adding a last class with the 97th percentile clearly visualizes and highlights the areas with excessively high values.The second classification is based on air quality guideline levels (AQG levels) established by the WHO and its interim targets. The interim targets are guidelines established for the pollutant in question that represent intermediate stepwise targets for reducing concentrations until compliance with the recommended maximum or guideline air quality levels is achieved.

According to Table 1, this classification consists of five ranges. The highest value of the interim objectives, set at 40 micrograms per cubic meter, also coincides with the critical annual limit set by the European Environment Agency (EEA). In 2024, the European Environment Agency lowered this limit value from 40 micrograms per cubic meter to 20 micrograms per cubic meter in line with the third WHO target [16].

### 3.2. Facilities Analysis

An analysis of the distribution of facilities according to the type of vulnerable population was carried out, following a methodology based on the geospatial analysis of the distribution of facilities proposed by Bosch-Checa [17].

In the initial phase, the geometry of the georeferenced facilities layer was converted to a standard format by applying vector geometry tools. Distance mapping was performed, generating a list of statistics. Next, a heat map was created by estimating the kernel density [18]. This process resulted in the distribution of the facilities in that area. In terms of symbolization, Jenks’ natural break-up method was chosen for its ability to identify and classify significant changes [19].

A weighting methodology was adopted to assess the distribution of installations, considering the importance assigned to each type of installation. The criteria determining this importance include the total time that each population group spend at each facility, the population concentration at each location, the range of air quality impacts on each population group, and their contextual situation in that environment. These weightings have been previously consulted on and discussed with experts in the context of the AVI Airluisa INNEST/2021/263 project.

The weighting leads to a higher risk as the assigned value increases. Facilities linked to sick people, in particular hospitals, receive the highest weighting. The table of assigned weightings shown in Table 2 was consulted on with experts, who adjusted and validated the weightings according to the criteria explained above. As a consequence, the layers of the four population types were merged, generating a global layer of facilities with their respective types and weights. 

The distribution analysis procedure, detailed in the previous section, was rerun to obtain a new heat map reflecting the distribution of facilities. Unlike the previous analysis, in this one, the weights attributed to the heat map’s weight field were incorporated to influence the final result.

### 3.3. Combined Analysis

The combined analysis phase aimed to evaluate environmental equity by integrating the data from both strands of the study. The objective was to assess the geospatial distribution of NO_2_ concentrations and facility types, taking into account their relative significance.

Raster maps from the NO_2_ concentration interpolation and facility heat map analyses were converted into vector polygons using specified intervals. These intervals corresponded to the highest inequality ranges, thereby focusing on areas with the greatest disparity. This approach facilitated the identification and delimitation of regions with significant levels of environmental inequity, combining insights from both strands of the analysis to provide a comprehensive understanding of the spatial distribution of air pollution and facility impacts on vulnerable populations.

## 4. Results and Discussion

The measurements collected by the sensors used in the investigation range from 12.225 µg/m^3^ (located at the green point in Figure 4) to the highest average of 89.95 µg/m^3^ recorded by the sensor at the entrance to CV-36, known as the Autovía de Torrent (red point in Figure 4). The two areas of most notable NO_2_ concentrations highlighted in Figure 4 coincide with two major vehicle entrances to the city, i.e., main roads or motorways.

Of the 99 sensors measuring NO_2_, more than half (53.54%) exceed the 40 µg/m^3^ annual critical value established by the WHO and the EEA. The remainder fall below this threshold, with 30.30% corresponding to the first WHO target (30–40 µg/m^3^), 11.11% to the second target (20–30 µg/m^3^), and a small portion to the third target (10–20 µg/m^3^). Notably, none of the annual averages reach the WHO AQG level of less than 10 µg/m^3^. These measurements are clearly reflected in Figure 5, revealing that most of the study area’s exhibited NO_2_ concentrations exceed the limit of 40 micrograms per cubic meter.

Regarding the identified areas of vulnerability concerning facilities, it is observed that the least vulnerable is the port region or Nazareth neighborhood, which is characterized by the scarcity of facilities related to the population groups studied. Similarly, the neighborhood of La Punta in the district of Quatre Carreres shows low vulnerability due to its composition of mostly market garden areas with few facilities. In contrast, the most vulnerable district is La Saïdia, including the extensive areas of Ciutat Vella, Extremurs, and El Pla del Reial, which are notable for their high presence of facilities, as evidenced in the heat maps in Figure 6. This result is explained by the weights assigned to facilities (Table 2) in these areas, highlighting their significant concentration.

Based on the percentile ranking of air quality, the first result of environmental inequity zones identifies a more critical zone in Ciutat de Valencia, covering approximately 0.12 km^2^ (“First result” in Figure 7). This area, located between the districts of Extremurs and Arrancapins, has residential buildings and several avenues and is particularly notable for its air quality concerns.

The second result focuses on areas exceeding the critical value of the World Health Organization (40 µg/m^3^). This result is divided into two larger zones compared to the first. The first zone, the largest at about 4.7 km^2^ (north area of “second result” in Figure 7), is located mainly in the districts of Extramural and La Saïdia, crosses the course of the river Túria in two segments, and contains the area of maximum vulnerability identified in the previous classification. The second zone of approximately 1.3 km^2^ (south-eastern area of “second result” in Figure 7), is located mainly in L’Eixample, the southern area of Pla del Real, and the western area of Camins al Grau, including 0.28 km^2^ of gardens along the course of the river Túria.

The following Figure 8 shows the total areas of the neighborhood affected by the second critical zone, which covers approximately 1.3 km^2^. The Table 3 specifies both the affected areas within each neighborhood and the percentage of the surface area of each neighborhood that is affected. These data are crucial to understand the geographical distribution of air pollution and the most vulnerable areas that require priority attention. The results show significant variations in the degree that neighborhoods are affected, indicating that air quality is not evenly distributed across the city. This detailed analysis identifies areas in need of urgent interventions to improve environmental health and reduce disparities in pollution exposure.

The most affected neighborhoods in terms of the percentage of area affected include SANT ANTONI, LA PETXINA, and TORMOS, each with 100% of their area affected. Other neighborhoods with high percentages of area affected are CAMI FONDO (91.72%), EL CALVARI (87.23%), and LA GRAN VIA (87.18%), These data highlight the severity of environmental inequity in certain urban areas, reflecting the need for specific interventions to improve air quality and reduce the vulnerability of the most affected communities.

On the other hand, some neighborhoods, such as EN CORTS, LA CREU DEL GRAU, and LA RAIOSA, show minimal impact, with less than 3% of their area affected. These differences in impact underline the uneven distribution of air pollution within the city and the importance of local approaches to public health and environmental policies.

## 5. Conclusions

The results of this methodology reveal significant potential beyond the city in which it was initially applied. Given that the methodology does not rely on any specific parameters, it can be applied to any urban area, regardless of its size or characteristics. The adaptability of the methodology is demonstrated by its ability to adjust the number of sensors needed to effectively cover the area under analysis. This flexibility ensures that the methodology is applicable to cities of different scales, making it a versatile tool for urban planning and environmental assessment.

The ability to analyze the most exposed areas through the annual amounts of nitrogen dioxide offers a valuable tool for decision-making, facilitating identification and action regarding environmental inequity areas. The versatility of this methodology extends to urban planning, as it allows for the distribution of facilities in cities, highlighting the least well-equipped areas. Furthermore, it stands out by identifying the areas with the highest risk, where the presence of vulnerable people to NO_2_ is combined with high concentrations of this pollutant, offering crucial data to enhance the quality of life in urban environments.

In the specific case of Valencia, the application of this methodology has provided further illuminating insights. It has been observed that the areas with the highest equity, both in terms of the amounts of nitrogen dioxide and the presence of NO_2_-vulnerable people, coincide mostly with areas of the city that lack vehicular traffic or are predominantly pedestrianized (streets in green in Figure 9). This correlation suggests that urban policies promoting sustainable mobility and traffic reduction not only contribute to improved air quality but also create safer and healthier environments for vulnerable groups. This demonstrates the potential for urban design to mitigate pollution impacts effectively.

These findings reinforce the notion that the application of this methodology not only facilitates the identification of areas with environmental inequity—where air quality is poorest—but also supports urban planning strategies that enhance residents’ health and well-being. In the context of Valencia, this comprehensive approach could aid in advancing initiatives to expand and strengthen pedestrian zones, encouraging the adoption of environmental and mobility policies that contribute to sustainability while mitigating the risks associated with air pollution.

## Figures and Tables

**Figure 1 ijerph-21-01278-f001:**
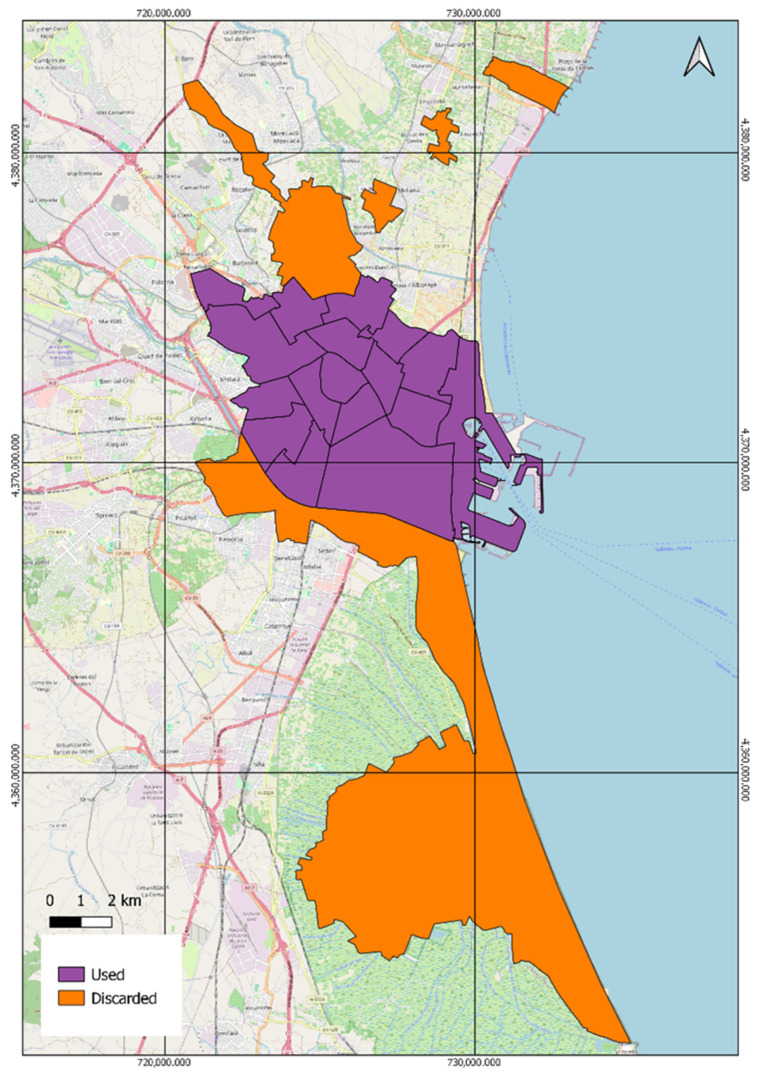
Used and discarded districts used in the study.

**Figure 2 ijerph-21-01278-f002:**
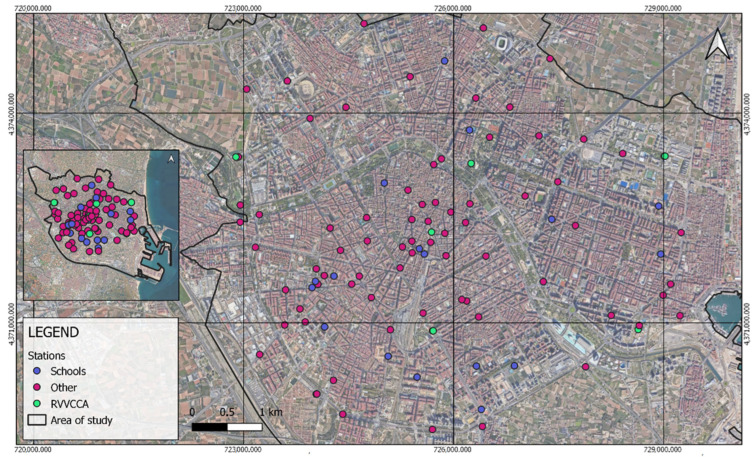
Location of sensor stations.

**Figure 3 ijerph-21-01278-f003:**
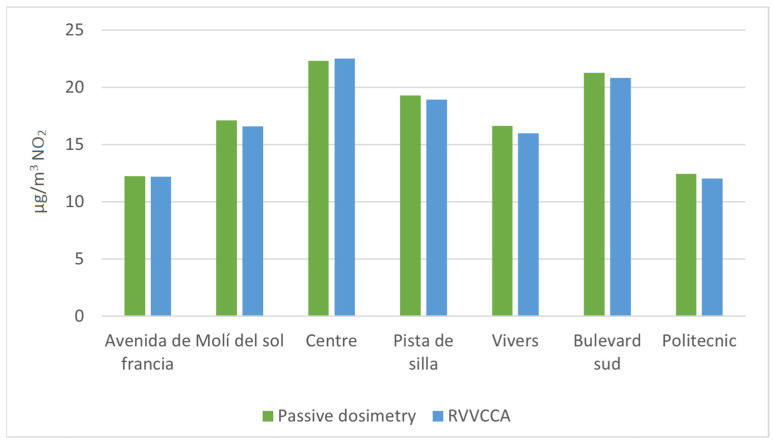
Comparison between the annual averages obtained from fixed air quality monitoring stations and annual averages from passive dosimetry.

**Figure 4 ijerph-21-01278-f004:**
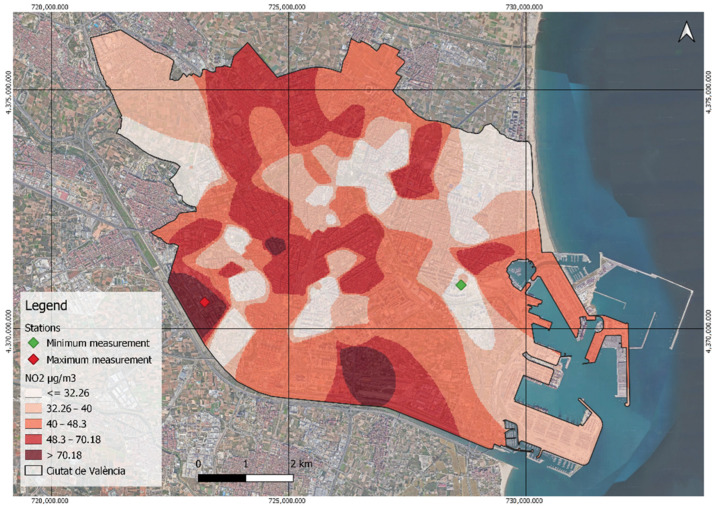
Classification of NO_2_ by percentiles and maximum and minimum measurements.

**Figure 5 ijerph-21-01278-f005:**
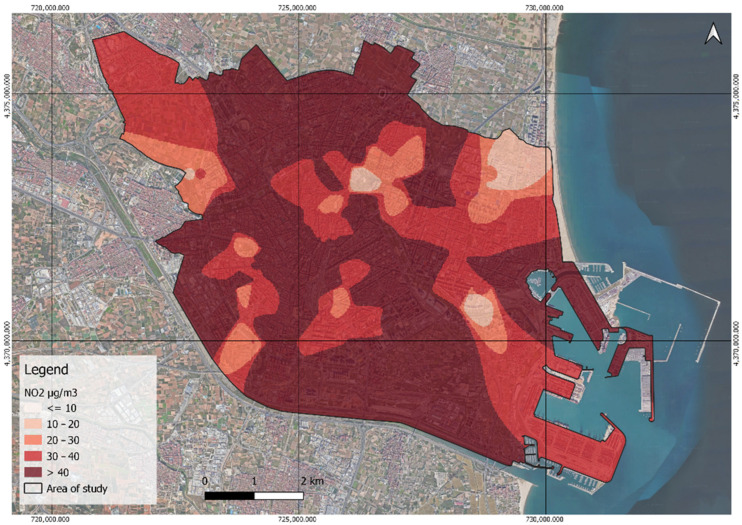
Classification of NO_2_ by interim targets and AQG levels.

**Figure 6 ijerph-21-01278-f006:**
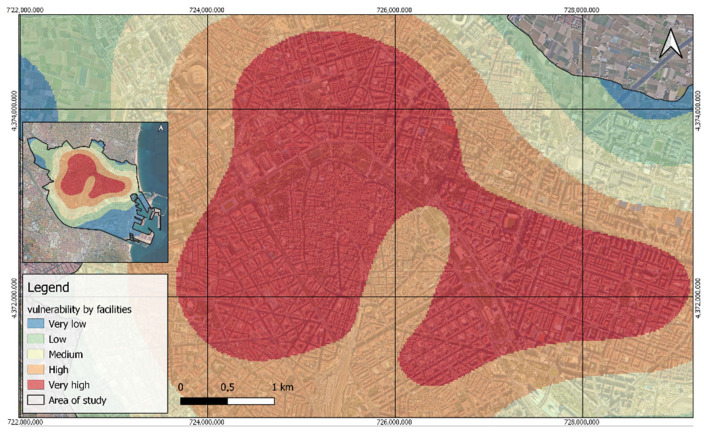
Heat map of the spatial distribution of weighted total facilities.

**Figure 7 ijerph-21-01278-f007:**
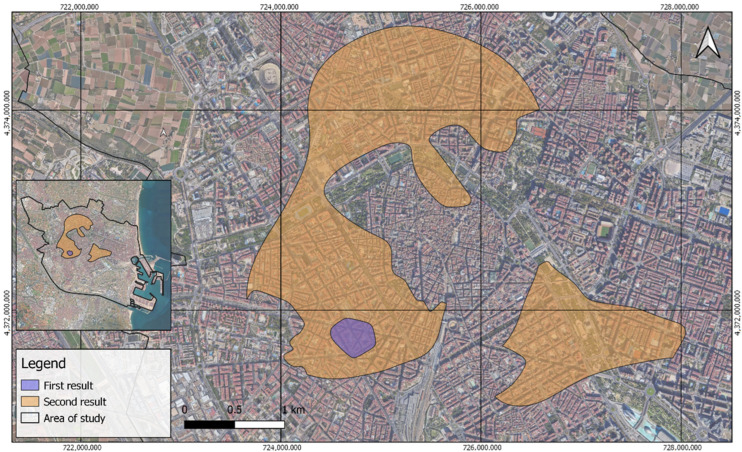
Key environmental equity outcomes.

**Figure 8 ijerph-21-01278-f008:**
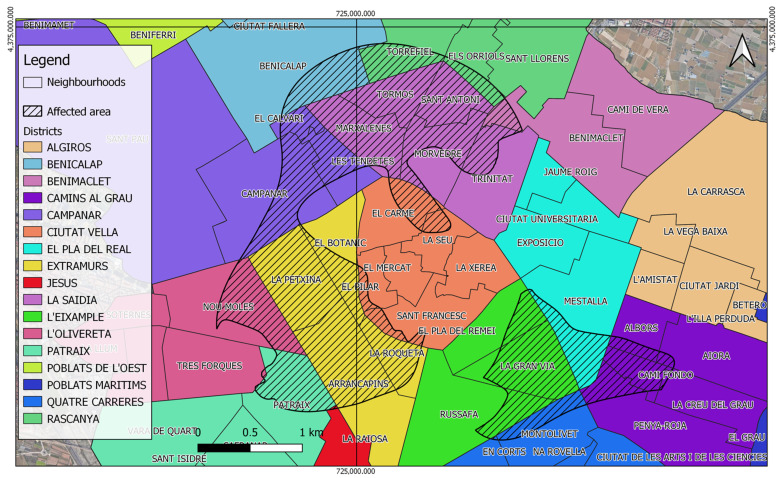
Districts, neighborhoods, and affected areas of Valencia.

**Figure 9 ijerph-21-01278-f009:**
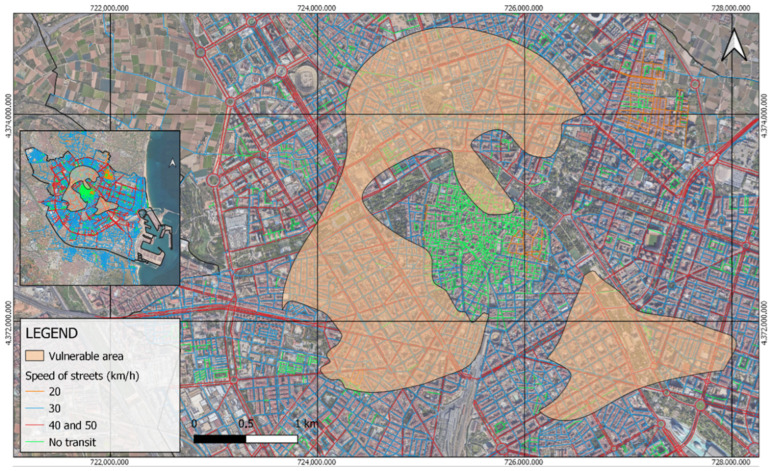
Vulnerable areas with speed of the streets.

**Table 1 ijerph-21-01278-t001:** Comparison limits, AQG levels (air quality guideline levels) and interim targets between WHO (World Health Organization) and EEA (European Environment Agency).

NO_2_ (µg/m^3^)	WHO	EEA
40	Interim target 1	Previous limit
30	Interim target 2	
20	Interim target 3	Limit 2024
10	AQG level	

**Table 2 ijerph-21-01278-t002:** Weightings assigned to each population group and to each type of facility.

Population Group	Type	Weighting
Children	Daycare centers and Early Childhood Education Centers	4
	Early Childhood Education Centers	3
	Primary and secondary schools	2
	Educational institutions and social centers	1
People over 65 years old	Residential homes	4
	Day centers	3
	Pensioners’ clubs and other meeting associations.	2
Chronically ill people	Hospitals	5
	Health centers	3
	Clinics	1
People who spend a significant amount of time outdoors	Sports facilities	1
	Children’s playgrounds	1

**Table 3 ijerph-21-01278-t003:** Table of affected areas in the second critical zone.

District	Neighborhood	Total Area (km^2^)	Affected Area (km^2^)	Percentage of Affected Area (%)
Ciutat vella	Sant francesc	0.438	0.059	13.47
	La seu	0.221	0.044	19.91
	El carme	0.384	0.126	32.81
	El pilar	0.162	0.036	22.22
Quatre carreres	En corts	0.364	0.001	0.27
	Montolivet	0.473	0.120	25.37
Camins al grau	Aiora	0.651	0.018	2.76
	Albors	0.256	0.054	21.09
	Cami fondo	0.157	0.144	91.72
	La creu del grau	0.391	0.008	2.05
	Penya-roja	0.911	0.115	12.62
Benimaclet	Benimaclet	0.743	0.062	8.34
Rascanya	Els orriols	0.397	0.077	19.40
	Torrefiel	0.700	0.186	26.57
Benicalap	Benicalap	1.718	0.258	15.02
L’eixample	El pla del remei	0.387	0.102	26.36
	Russafa	0.877	0.140	15.96
	La gran via	0.468	0.408	87.18
Extramurs	El botanic	0.370	0.157	42.43
	Arrancapins	0.874	0.520	59.50
	La petxina	0.496	0.496	100.00
	La roqueta	0.230	0.182	79.13
Campanar	El calvari	0.094	0.082	87.23
	Les tendetes	0.258	0.161	62.40
	Campanar	0.987	0.346	35.06
La saidia	Marxalenes	0.390	0.379	97.18
	Morvedre	0.428	0.260	60.75
	Sant antoni	0.250	0.250	100.00
	Tormos	0.280	0.280	100.00
	Trinitat	0.595	0.130	21.85
L’olivereta	Nou moles	0.739	0.339	45.87
	Tres forques	0.527	0.003	0.57
Patraix	Patraix	0.621	0.286	46.05
El pla del real	Mestalla	0.843	0.228	27.05
Jesus	La raiosa	0.375	0.002	0.53

## Data Availability

The data presented in this study are available on request from the corresponding author. The data used in this study are currently undergoing updates and refinements, and have not yet been of-ficially published. As the analysis is still ongoing, and to ensure the accuracy and consistency of the final results, the data are not publicly available at this stage. However, the data may be provided upon request to researchers interested in further exploring this work, once the updating process is completed.

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
