# Peer review of "Air Quality Geospatial Analysis in Vulnerable Areas. Case Study of Valencia (Spain)"

_ijerph, 2024, doi:10.3390/ijerph21101278_

Round 1

Reviewer 1 Report

Comments and Suggestions for Authors

1. Please ensure the accurate and consistent use of subscripts when presenting molecular formulas of airborne pollutants in your citations. For instance, the '2' in nitrogen dioxide (NO2) should be subscripted to accurately denote the presence of two oxygen atoms. I recommend a thorough review of all molecular formulas throughout the paper to confirm their adherence to standard scientific notation.

2.            The introduction briefly references previous studies on air pollution and health impacts but does not provide a comprehensive review of relevant literature. There is limited discussion on how this study builds upon or differs from existing research. A more detailed literature review would help situate the study within the broader field and justify its significance.

3.            The choice of Valencia as the study area is mentioned but not thoroughly justified. Why is Valencia particularly relevant or representative for this type of study? Are there unique characteristics of Valencia’s urban environment or demographic composition that make it a critical case for analyzing NOâ‚‚ concentrations and vulnerable populations?

4.            Can the authors clearly state the specific research questions or hypotheses that guide the study? How do these objectives align with the broader context of air quality and public health research?

5.            The analysis is limited to the year 2022. How might the findings differ if a multi-year dataset were used? Are there any indications that 2022 was an atypical year in terms of NOâ‚‚ levels or other environmental factors?

6.            The article mentions the use of Inverse Distance Weighting (IDW) for interpolation but lacks detailed parameters and justification for this choice. Why was IDW chosen over other interpolation methods.

7.            The study focuses on NOâ‚‚ concentrations without extensively discussing the direct health impacts on the population. How do the authors plan to establish a more direct link between NOâ‚‚ levels and specific health outcomes in vulnerable populations?

8.            The study is localized to Valencia. How applicable are the findings to other urban areas with similar or different characteristics? Can the methodology and conclusions be generalized beyond this specific case study?

Comments on the Quality of English Language

Language needs to be improved. 

Author Response

Reviewer 1 

  1. Please ensure the accurate and consistent use of subscripts when presenting molecular formulas of  airborne pollutants in your citations. For instance, the '2' in nitrogen dioxide (NO2) should be  subscripted to accurately denote the presence of two oxygen atoms. I recommend a thorough review  of all molecular formulas throughout the paper to confirm their adherence to standard scientific  notation.

Response 1: thank you very much for the note, it has been corrected throughout the text.

  1. The introduction briefly references previous studies on air pollution and health impacts but does  not provide a comprehensive review of relevant literature. There is limited discussion on how this  study builds upon or differs from existing research. A more detailed literature review would help  situate the study within the broader field and justify its significance. 

Response 2: thank you very much for the note, the introduction has been made and the paragraph lines 74-88 on page 1 has been added for correction.

  1. The choice of Valencia as the study area is mentioned but not thoroughly justified. Why is Valencia  particularly relevant or representative for this type of study? Are there unique characteristics of  Valencia’s urban environment or demographic composition that make it a critical case for analyzing  NOâ‚‚ concentrations and vulnerable populations? 

Response 3: thank you very much for the observation, I hope that the editing of the text in rows 110-122 on page 3 justifies the choice a bit more.

  1. Can the authors clearly state the specific research questions or hypotheses that guide the study?  How do these objectives align with the broader context of air quality and public health research? 

Response 4: the objectives of this study align with the broader context of air quality research by addressing the improvement of air quality through the geolocation of hotspots based on NOâ‚‚ concentrations, allowing for the implementation of specific corrective measures. This research is directly related to public health, as it focuses on geolocating areas with high concentrations that could affect the health of people most vulnerable to this gas, providing crucial information to protect vulnerable populations and improve their living conditions.

  1. The analysis is limited to the year 2022. How might the findings differ if a multi-year dataset were  used? Are there any indications that 2022 was an atypical year in terms of NOâ‚‚ levels or other  environmental factors? 

Response 5: Thank you for pointing but the data collection was specifically conducted in 2022, which is why the analysis is based on that year's data.

  1. The article mentions the use of Inverse Distance Weighting (IDW) for interpolation but lacks  detailed parameters and justification for this choice. Why was IDW chosen over other interpolation  methods. 

Response 6: Thank you for pointing I hope that the editing of the text in rows 218-233 on page 6 justifies the lack details

  1. The study focuses on NOâ‚‚ concentrations without extensively discussing the direct health impacts  on the population. How do the authors plan to establish a more direct link between NOâ‚‚ levels and  specific health outcomes in vulnerable populations? 

Response 7: we have added text in the introduction to rectify it

  1. The study is localized to Valencia. How applicable are the findings to other urban areas with similar  or different characteristics? Can the methodology and conclusions be generalized beyond this specific  case study? 

Response 8:Thank you for pointing I hope that the adding text in rows 366-371 on page 14 justifies that this methodology can be generalized to other cases

- Language needs to be improved. 

there are changes in phrases and expressions throughout the text, also marked in red for the improvement of the English language.

- should add more relevant references.

Reviewer 2 Report

Comments and Suggestions for Authors

The manuscript entitled "Air Quality Geospatial Analysis in Vulnerable Areas: Case Study of Valencia (Spain)" aims to provide evidence of environmental inequity affecting vulnerable groups by analyzing NO2 concentration maps and maps of vulnerability levels. However, the manuscript is poorly written, making it difficult for readers to understand. I suggest the authors focus on the following issues:

1. Clearly state the research objectives to provide better direction and focus for the study.

2. In section 2.1, include references that highlight high NO2 levels in Valencia to support the study's context.

3. Provide more concise and detailed information in the methodology section to improve reliability. This should include:

-          Justification for weighting schemes

-          Brand/model and reliability level of the sensors used

-          Approach to minimize meteorological biases

-          Improvement of the combined analysis in section 4.3

4. NO2 levels are highly dependent on seasonal changes. Clarify how the data accounts for this, especially considering that each sampling point was measured four times a year for 14 days each.

5. Explain the objectives of the NO2 measurement campaign and identify who was involved. Since dosimeters may be attached to individuals, their occupations should be considered as they might influence NO2 exposure levels.

6.: Include a map showing all districts mentioned (e.g., Nazareth, La Punta, La Saidia, Ciutat) to help readers unfamiliar with the Spanish context.

7. Ensure that the connection between Figure 5 and Table 2 is clear and logical.

8. Provide a clear explanation of Figure 6 and how to interpret it.

9. Ensure that the conclusions are directly drawn from the research findings and clearly presented.

Comments on the Quality of English Language

Consult with native English speakers experienced in research publication to improve clarity and readability.

Author Response

Reviewer 2 

  1. Clearly state the research objectives to provide better direction and focus for the study. 

Response 1: thank you very much for the observation, I hope that the editing of the text in rows 75-89 on page 2 justifies the objective a bit more.

  1. In section 2.1, include references that highlight high NO2 levels in Valencia to support the study's  context. 

Response 2: thank you very much for the observation, I hope that the editing of the text in rows 101-105 on page 3 justifies the objective a bit more.

  1. Provide more concise and detailed information in the methodology section to improve reliability.  This should include: Justification for weighting schemes . Brand/model and reliability level of the sensors used. Approach to minimize meteorological biases. Improvement of the combined analysis in section 4.3.

Response 3: thank you very much for the observation, I hope that the editing of the text in rows 124-139 on page 4 justifies the reliability level of the sensors used, new text in rows 142-144 on page 4  clear the approach to minimise meteorological biases and rowa 263-272 on page 7 improves the combined analysis.

  1. NO2 levels are highly dependent on seasonal changes. Clarify how the data accounts for this,  especially considering that each sampling point was measured four times a year for 14 days each. 

Response 3: thank you very much for the observation, I hope that the editing of the text in rows 148-152 on page 4 clarifies it

  1. Explain the objectives of the NO2 measurement campaign and identify who was involved. Since  dosimeters may be attached to individuals, their occupations should be considered as they might  influence NO2 exposure levels. 

  1. Include a map showing all districts mentioned (e.g., Nazareth, La Punta, La Saidia, Ciutat) to help  readers unfamiliar with the Spanish context. 

Response 8: thank you for your appointment, I hope that the editing othe text, the new table and the new map in rows 320-341 on pages 12-13 clarifies it

  1. Ensure that the connection between Figure 5 and Table 2 is clear and logical.

Response 8: thank you for your appointment, I hope that the editing of the text in rows 299-301 on page 9 clarifies it

  1. Provide a clear explanation of Figure 6 and how to interpret it. 

Response 8: thank you for your appointment, I hope that the editing of the text in rows 309-317 on page 9 clarifies it

  1. Ensure that the conclusions are directly drawn from the research findings and clearly presented. 

Response 8: thank you for your appointment, I just add son rows to explain better the conclusions ( rows 362-368, page 14)

- Consult with native English speakers experienced in research publication to improve clarity and  readability.

there are changes in phrases and expressions throughout the text, also marked in red for the improvement of the English language.

Round 2

Reviewer 1 Report

Comments and Suggestions for Authors

·       Spell out the full term first, then use the acronym for the rest of the article. There’s no need to repeat it every time.

·       Is it 4.3 or 3.3?  what kind of intervals did you use, please specify.

·       Section 4, Table 3: why they all are in capital letters?

·       Some references are not in the correct format. Please be consistent and check references  again,

Comments on the Quality of English Language

can be improved.

Author Response

Commet 1:Is it 4.3 or 3.3?  what kind of intervals did you use, please specify.

Response 1: Thank you for pointing this out. You are sure is 3.3, It is already rectified

Coment2: Section 4, Table 3: why they all are in capital letters?

Response 2: Thank you for pointing this out. It is already rectified, they are now in lowercase except the first letter.

Coment 3: Some references are not in the correct format. Please be consistent and check references  again

Response 3:thank you very much, the last reference was left over, the rest is done automatically with the Mendeley software so there shouldn't be any other errors

Reviewer 2 Report

Comments and Suggestions for Authors

The revised manuscript titled "Air Quality Geospatial Analysis in Vulnerable Areas: A Case Study of Valencia (Spain)" successfully clarifies the study's objectives. The inclusion of Table 3, which details vulnerable areas, is excellent. However, several concerns remain unaddressed, particularly in the methodology and results/discussion sections.

As outlined in the introduction, the authors aim to develop a methodology for identifying vulnerable zones that can be applied to any urban city. Therefore, the methodology in this study must be clearly articulated, with an evaluation of its effectiveness. I recommend that the authors expand on the following methodological aspects:

1. The manuscript mentions the use of 'Palmes diffusion tubes.' The authors should also specify the brand/model of the sensors, the absorptive solution used, and the chemical analysis method employed for NOâ‚‚ analysis. Additionally, the reliability of these methods should be reported.

2. The methodology section states that measurements were conducted four times at each location, covering different seasons, with each measurement lasting 14 days. However, the authors do not discuss the seasonal variations in NOâ‚‚ levels. This aspect should be addressed in the discussion.

3. In Figure 2, the monitoring stations should be labeled to correspond with all vulnerable facilities mentioned in Section 2.3.

4. It is unclear whether the second classification uses the WHO Interim Target 1 threshold. This should be clarified.

5. The combined analysis in Section 4.3 requires significant improvement. The process of integrating the NOâ‚‚ level map with the weighting scheme map is not clearly explained. 

Results and Discussion

6. Figure 3 shows that the peak NOâ‚‚ level is observed in areas with white roofs. What is it? Although the authors discuss the association with vehicles and road activities, they do not mention the role of white-roof land use activities. This omission should be addressed.

7. The weighted total facilities in Figure 5 should correspond to a scale of 1-5, as shown in Table 2, rather than qualitative levels.

8. Figure 6 is confusing and does not seem to correspond with the NOâ‚‚ levels in Figure 3. The distinction between the first and second results is unclear. It should be clarified whether the first result is based on the 97th percentile NOâ‚‚ level and the second result on the WHO Interim Target 1. This clarification should be reflected both in Figure 6 and the accompanying text.

Author Response

comment 1: The manuscript mentions the use of 'Palmes diffusion tubes.' The authors should also specify the brand/model of the sensors, the absorptive solution used, and the chemical analysis method employed for NOâ‚‚ analysis. Additionally, the reliability of these methods should be reported.

Response 1: rows 145-146 and 166-172

Comment 2: The methodology section states that measurements were conducted four times at each location, covering different seasons, with each measurement lasting 14 days. However, the authors do not discuss the seasonal variations in NOâ‚‚ levels. This aspect should be addressed in the discussion.

Response 2: thank you very much for the observation, I hope that the editing of the text in rows 149-159 and 172-176 on page 3 justifies it a bit more.

Comment 3:  In Figure 2, the monitoring stations should be labeled to correspond with all vulnerable facilities mentioned in Section 2.3.

Response 3: Thank you very much for your insightful comment. We appreciate your suggestion to label the monitoring stations in Figure 2 to correspond with all vulnerable facilities mentioned in Section 2.3. However, we believe that labeling each of the schools directly in the figure might result in an overcrowded image, making it difficult to interpret. To address this, we have added an explanatory sentence in lines 161-163, which clarifies that these facilities are considered and utilized in the subsequent section. This adjustment ensures that the connection between the monitoring stations and the vulnerable facilities is clear without overcomplicating the figure itself.

Comment 4:  It is unclear whether the second classification uses the WHO Interim Target 1 threshold. This should be clarified.

Response 4: thank you very much for the observation, I hope that the editing of the text in rows 244-248 on page 3 clarifies it a bit more 

comment 5:  The combined analysis in Section 4.3 requires significant improvement. The process of integrating the NOâ‚‚ level map with the weighting scheme map is not clearly explained. 

Response 5: thank you very much for the note, the whole section d4.3 combined analysis has been modified to be clearer.

Comment 6: . Figure 3 shows that the peak NOâ‚‚ level is observed in areas with white roofs. What is it? Although the authors discuss the association with vehicles and road activities, they do not mention the role of white-roof land use activities. This omission should be addressed.

Response 6: Figure 3 shows that the maximum measurement point is in the south-west area (marked with a red diamond) and the minimum measurement point is in the east (marked with a green diamond) in an area with a white roof. In the case of the lower measurements, they do not coincide with any specific land use as they are found in areas of buildings, crops, the harbour, etc.

Comment 7: The weighted total facilities in Figure 5 should correspond to a scale of 1-5, as shown in Table 2, rather than qualitative levels.

Response 7: thank you very much for the note, figure 5 has been taken into account and changed.

Comment 8: Figure 6 is confusing and does not seem to correspond with the NOâ‚‚ levels in Figure 3. The distinction between the first and second results is unclear. It should be clarified whether the first result is based on the 97th percentile NOâ‚‚ level and the second result on the WHO Interim Target 1. This clarification should be reflected both in Figure 6 and the accompanying text.

Response 8: Thank you very much for your comment, figure 3 and 6 do correspond to each other, the change is that in figure 6, the scale is enlarged to have more detail of the result areas, the legend of figure 6 has been changed to understand where each result comes from.